# Genome-wide association study of cassava starch paste properties

**Cristiano Silva dos Santos**[1☯], **Massaine Bandeira Sousa**[1☯], **Ana Carla Brito**[2‡], **Luciana Alves de Oliveira**[2‡], **Carlos Wanderlei Piler Carvalho**[3‡], **Eder Jorge de Oliveira**[2☯]*

**1** Universidade Federal do Recôncavo da Bahia, Centro de Ciências Agrárias, Ambientais e Biológicas, Rua Rui Barbosa, Cruz das Almas, BA, Brazil, **2** Embrapa Mandioca e Fruticultura, Rua da Embrapa, Cruz das Almas, BA, Brazil, **3** Embrapa Agroindústria de Alimentos, Avenida das Américas, Guaratiba, RJ, Brazil

☯ These authors contributed equally to this work.
‡ ACB, LADO and CWPC also contributed equally to this work.
* eder.oliveira@embrapa.br

**Data Availability Statement:** All datasets generated for this study can be found in the article, supplementary material, and Figshare (https://doi.org/10.6084/m9.figshare.14773005.v1).

## Abstract

An understanding of cassava starch paste properties (CSPP) can contribute to the selection of clones with differentiated starches. This study aimed to identify genomic regions associated with CSPP using different genome-wide association study (GWAS) methods (MLM, MLMM, and Farm-CPU). The GWAS was performed using 23,078 single-nucleotide polymorphisms (SNPs). The rapid viscoanalyzer (RVA) parameters were pasting temperature (PastTemp), peak viscosity (PeakVisc), hot-paste viscosity (Hot-PVisc), cool-paste viscosity (Cold-PVisc), final viscosity (FinalVis), breakdown (BreDow), and setback (Setback). Broad phenotypic and molecular diversity was identified based on the genomic kinship matrix. The broad-sense heritability estimates ($h^2$) ranged from moderate to high magnitudes (0.66 to 0.76). The linkage disequilibrium (LD) declined to between 0.3 and 2.0 Mb ($r^2$ <0.1) for most chromosomes, except chromosome 17, which exhibited an extensive LD. Thirteen SNPs were found to be significantly associated with CSPP, on chromosomes 3, 8, 17, and 18. Only the BreDow trait had no associated SNPs. The regional marker-trait associations on chromosome 18 indicate a LD block between 2907312 and 3567816 bp and that SNP S18_3081635 was associated with SetBack, FinalVis, and Cold-PVisc (all three GWAS methods) and with Hot-PVisc (MLM), indicating that this SNP can track these four traits simultaneously. The variance explained by the SNPs ranged from 0.13 to 0.18 for Set-Back, FinalVis, and Cold-PVisc and from 0.06 to 0.09 for PeakVisc and Hot-PVisc. The results indicated additive effects of the genetic control of Cold-PVisc, FinalVis, Hot-PVisc, and SetBack, especially on the large LD block on chromosome 18. One transcript encoding the glycosyl hydrolase family 35 enzymes on chromosome 17 and one encoding the mannose-p-dolichol utilization defect 1 protein on chromosome 18 were the most likely candidate genes for the regulation of CSPP. These results underline the potential for the assisted selection of high-value starches to improve cassava root quality through breeding programs.

**Funding:** • Eder Jorge de Oliveira: CNPq (Conselho Nacional de Desenvolvimento Científico e Tecnológico). Grant number: 409229/2018-0, 442050/2019-4 and 303912/2018-9 • Eder Jorge de Oliveira: FAPESB (Fundação de Amparo à Pesquisa do Estado da Bahia). Grant number: Pronem 15/2014 • Ana Carla Brito: FAPESB (Fundação de Amparo à Pesquisa do Estado da Bahia). Grant number: DCR0005/2015 • Eder Jorge de Oliveira: UK's Foreign, Commonwealth & Development Office (FCDO) and the Bill & Melinda Gates Foundation. Grant number: INV-007637. Under the grant conditions of the Foundation, a Creative Commons Attribution 4.0 Generic License has already been assigned to the Author Accepted Manuscript version that might arise from this submission. • The funder provided support in the form of fellowship and funds for the research, but did not have any additional role in the study design, data collection and analysis, decision to publish, or preparation of the manuscript.

**Competing interests:** The authors declare that the research was conducted in the absence of any commercial or financial relationships that could be construed as a potential conflict of interest.

# 1 Introduction

Cassava (*Manihot esculenta* Crantz) is one of the most important species globally for starch production. In addition, it is a staple food for approximately 800 million people in tropical countries of Latin America, Asia and Africa [1], and is considered to be of strategic importance as a subsistence crop for food security, especially in several African countries [2]. In addition to its being consumed fresh, cooked, fried, or in the preparation of typical dishes, cassava is widely used in the food industry more broadly, mainly in the form of dehydrated chips, cassava flour, and starch [3]. Cassava starch has unique physicochemical characteristics, such as a high swelling capacity, high peak viscosity, and low retrogradation tendency [4]. Starch retrogradation is generally considered to have undesirable effects on food products, but is desirable in some applications, such as in the production of breakfast cereals, dehydrated purees, and resistant starches, which have several nutritional properties [5]. This process in which disaggregate the starch components, amylose and amylopectin chains, in a gelatinized starch paste and reassociate to form more ordered structures. Amylose and amylopectin are two types of glucose polymers presents in starch, in which amylose, takes the form of a predominantly linear chain, and amylopectin, which is highly branched and contains many short-chain clusters. Starch granules are generally composed of an amorphous region interspersed with concentric semicrystalline rings, which exhibit alternating amorphous and crystalline materials [6, 7]

Starches from different crops exhibit great variation in their properties, which can be amplified by physical (pregelatinization, hydrothermal, and non-thermal processes) and chemical modifications (oxidation, esterification, and etherification) [8]. Native cassava starch has a lesser retrogradation tendency than cereal and potato starches. The different forms (round, oval, polyhedral), sizes (3 to 110 μm), and especially amylose and amylopectin contents of starches provide significant differences in their properties. In addition, starch properties can be altered by the presence of other components, such as proteins and lipids [9].

The pasting and gelatinization properties of starch are important because they define its industrial applications and have a strong impact on the quality of the final product [10]. In addition, current changes in consumers' eating habits have led to a preference for less-processed products, thus increasing the demand for native starches with specific profiles for industrial use.

The improvement of pasting properties has become a key objective of cassava breeding programs [11, 12]. Genetic studies of starches, especially of QTLs (quantitative trait loci) that control pasting properties and their effects on food quality, have been performed to understand the genetic basis of these quantitative traits. However, these mapping efforts rely exclusively on related genotypes, examining only one small panel of the full genetic diversity of cassava germplasm [11, 13]. After developing a thorough understanding of the genetic basis of starch pasting properties, the identification of QTLs associated with these characteristics may contribute to marker-assisted selection (MAS), so that new strategies can be applied to increase the selection efficiency and genetic gain from these starch traits [11, 14].

Currently, the increased availability of genomic resources has made it possible to analyze genotypic data and use them to better understand phenotypic variation. These results come from next-generation DNA sequencing (NGS) technologies, such as genotyping by sequencing (GBS), which enables the use of genome-wide association studies (GWAS) to more effectively investigate the genetic diversity of cassava germplasm. GWAS is a powerful method to identify genomic regions associated with the phenotypic variation for use in the selection of clones with traits of interest [13, 15–18]

GWAS makes it possible to choose parents for QTL analyses, and therefore is a complementary strategy for more refined mapping of quantitative traits. GWAS analysis platforms

have enabled the rapid processing of large phenotypic datasets that take into account different environments, years of cultivation, treatments, and replicates, improving the resolution of QTL mapping [19].

GWAS has been used to investigate the genetic architecture and mechanisms related to complex agronomic traits, such as rice grain production [20], corn starch content [21], and plant architecture of sorghum [22]. In cassava, GWAS has been used to understand and locate genes associated with dry matter content in the roots and carotenoid content [15, 23], as well as greater resistance to cassava mosaic virus [17, 23], cassava root rot [18], and cyanide content [24]. However, no GWAS approach has yet been used to explore genetic basis of cassava starch paste properties.

Although the genetic diversity of cassava exhibits many different natural phenotypes that can be exploited in the breeding of the species, molecular advances are still limited [18]. Thus, it is necessary to identify the gene regions associated with traits of greater interest in order to contribute to the selection of elite plants in breeding programs. Therefore, the present study investigated the effects of molecular polymorphisms associated with cassava starch paste properties using GWAS. We evaluated a diverse set of cassava accessions to provide a comprehensive view of the population structure and diversity of the species, their effects on GWAS, as well as the dynamics of linkage disequilibrium in different chromosomes, and finally to identify and locate genomic regions associated with the pasting properties of cassava starch.

## 2 Material and methods

### 2.1 Plant material

The germplasm consisted of 876 cassava accessions belonging to the Cassava Germplasm Bank (CGB) of Embrapa Mandioca e Fruticultura (Embrapa Cassava and Fruits), which is located in Cruz das Almas, Bahia, Brazil (12˚40'19"S, 39˚06'22" W, 226 m altitude). This germplasm is composed of landraces identified by farmers or research institutions as well as improved varieties originating from different Brazilian states and other countries (e.g., Colombia, Mexico, Venezuela, Panama, and Nigeria) (S1 Table).

Planting occurred in 2015 and 2016 at the beginning of the rainy season in the "Recôncavo da Bahia" region (May–July). Stakes of 15 to 20 cm were planted at a spacing of 0.9 m between rows and 0.8 m between plants. The experimental design was an augmented block design with 12 replicates and plots with 16 plants (two lines with 8 plants each).

The planting region is located in a geomorphological unit known as a coastal board, which presents characteristics such as deep soils, flat or slightly undulating topography, and the occurrence of cohesive horizons in some soils. It has a warm and humid tropical climate (Tropical Wet Savannah (Aw) to Ttropical Monsoon (Am) according to the Köppen classification) with an average annual rainfall of 1,170 mm (varying from 900 to 1,300 mm) and an average annual temperature of 24.1˚C. During the plant growth period, March to August was the rainiest, while September to February was the driest (S2 Table).

### 2.2 Starch extraction from cassava roots

The analyzed roots were selected during the harvesting process based on their commercial pattern (i.e., no mechanical damage). Subsequently, these roots were transported to the laboratory, washed with running water to remove impurities from the soil, and then processed. The root pulp was sectioned into cubes, with at least 2 kg of root material processed per accession. The roots were ground in a blender with a non-cutting propeller for 1 minute at a root-to-cold water ratio of 1:1. This process was repeated once after a 1-minute break.

For particle size analysis, the ground root mixture was filtered through voile-type fabric placed on a sieve (100 μm) over a 5-liter plastic bucket. Thereafter, the slurry was washed with 3.5 L of cold water for starch extraction. The filtrate was then placed in a cold room at 5˚C for 12 hours for decantation of the starch. After this time, the supernatant was discarded and the decanted starch at the bottom of the vessel was washed with 20 mL of 95% ethyl alcohol to accelerate the drying process. The alcohol was then discarded. Then, the starch was transferred to aluminum trays and conditioned in an oven with forced air circulation at 40˚C until completely dry (10–14% moisture). The dry starch was gently macerated with the aid of a mortar and pestle to obtain a fine powder. The starch was then packed in plastic bags, vacuum-sealed, and stored at 5˚C for further analysis.

## 2.3 Pasting properties analysis

Cassava starch pasting properties were evaluated using a rapid visco analyzer (RVA, model RVA-4500 and series 4, NewPort Scientific, Sydney, Australia) using the standard 1 configuration of Thermocline software for Windows, version 3. Three grams of starch from each cassava accession (14% moisture, wet basis) and 25 g of distilled water were directly weighed in the RVA's aluminum vessel. The sample and water weight corrections required to obtain 14% moisture were provided by the aforementioned software. The starch and water added to the vessel were mixed using the propeller coupled to the RVA. The starch suspension was then subjected to the following schedule (temperature/time): 50˚C for 1 minute; heating from 50 to 95˚C (6˚C/min); maintenance of the paste at 95˚C for 2.5 minutes; cooling from 95 to 50˚C (6˚C/minute) and maintenance of the paste at 50˚C for 2 minutes. The suspension was then shaken at 160 rpm throughout the analysis, which was performed twice.

Paste viscosity was expressed in centipoise (cP) and temperature was expressed in˚C. The total analysis time was 13 min for each duplicate. During this period, the following characteristics were evaluated: pasting temperature (PastTemp), peak viscosity (PeakVisc), hot paste viscosity (Hot-PVisc), cool paste viscosity at 50˚C (Cold-PVisc), final viscosity (FinalVis), breakdown (BreDow, which is the difference between PeakVisc and Hot-Pvisc), and setback (SetBack, which is the retrogradation tendency or difference between Cold-PVisc and Hot-Pvisc).

## 2.4 Data analysis

Statistical analyses were performed in R software version 4.0.1 [25] unless otherwise noted. The BLUPs for each of the pasting properties were estimated using the lme4 package from R based on the following mixed linear model: y = Xu+Zg+e, where y is the vector of phenotypic data, u is the general mean (fixed effect), g is the vector of the genotypic effects of the accessions (assumed as random), e is the vector of errors or residuals (random), and X and Z represent the incidence matrices for the respective effects of u and g of the mixed model. The lme4 package was also used to estimate the variance components obtained by the restricted maximum likelihood (REML) and broad-sense heritability ($h^2$). The $h^2$ values of the traits were estimated according to $h^2 = \frac{\sigma_g^2}{\sigma_g^2 + \sigma_e^2}$, where $\sigma_g^2$ is the genotypic variance and $\sigma_e^2$ is the residual variance.

## 2.5 DNA extraction

DNA was extracted from young leaves according to the CTAB (cetyltrimethylammonium bromide) protocol as described [26] with few modifications, being the addition of polyvinylpyrrolidone (PVP) and increasing the concentration of 2-mercaptoethanol to 0.4%, to break down

organic material, especially tannins and phenolics. DNA quality was assessed by 1.0% (w/v) agarose gel quantitation stained with ethidium bromide (1.0 mg/L) in 0.5 x TBE buffer (45 mM Tris-borate, 1 mM EDTA and distilled water qsp), visualized in UV light and recorded with the Gel Logic 212 Pro Photodocument (Carestream Molecular Imaging, New Haven, USA) by visual comparison with a number of DNA concentrations known from Lambda phage (Invitrogen, Carlsbad, CA). The DNA was diluted in TE buffer (10 mM Tris-HCl and 1 mM EDTA) to a final concentration of 60 ng/μl, and the quality was checked by digestion of 250 ng of the genomic DNA of 10 random samples with *Eco*RI (New England Biolabs, Boston, USA) at 65°C for two hours followed by visualization on agarose gel.

## 2.6 Genotyping-by-sequencing (GBS)

DNA samples were genotyped at the Genomic Diversity Facility of Cornell University (http://www.biotech.cornell.edu/brc/genomic-diversity-facility). The basic protocol for GBS has been described by [27]. As recommended by [28], the DNA was digested by the enzyme *Ape*KI, which is a type II restriction endonuclease that recognizes a 5-base degenerate sequence (GCWGC, where W is A or T) in lengths of 100 bp. Binding between the *Ape*KI cut-off fragments and the adapter was performed after the digestion of the samples and implementation of a multiplex system with 192 samples for sequencing. GBS was performed using a Genome Analyzer 2000 (Illumina, Inc., San Diego, CA).

## 2.7 Association between molecular polymorphisms and pasting properties

The SNPs were subjected to a quality filter in TASSEL software [29], where the tags with more than 20% missing data and those with an MAF < 0.05% were discarded. To test the association between pasting properties and gene sequences, the Genome Association and Prediction Integrated Tool (GAPIT) 3.0 package [30] for R was used. We implemented a mixed linear model (MLM), a multi-locus mixed model (MLMM), and a fixed and random model for circulating probability unification (FarmCPU) methods in the GAPIT 3.0 package.

The MLM notation was $y = s_i + Q + K + e$, where y is the vector of phenotypic observations, $Q$ is the population structure matrix normally represented by proportions of individuals belonging to subpopulations or by principal components (PC) from genetic markers, $K$ is the kinship matrix that defines the relationship among individuals, and $e$ is the random vector of residual effects. Normally, $Q$ and $K$ are fitted as fixed effects. The MLM is a single locus analysis that uses $Q$ and $K$ in the model to reduce false positives from family relatedness and population structure.

The MLMM is a multi-locus analysis in which $y = s_i + S + Q + K + e$, where $S$ are pseudo quantitative trait nucleotides (QTN) fitted by testing markers in a stepwise MLM as additional covariates for the marker test. The covariates are adjusted through forward and backward stepwise regression of the mixed model. FarmCPU is also a multi-locus analysis given by the formula: $y = s_i + S + e$. The latter method incorporates multiple markers as covariates in a stepwise MLM to partially remove the confounding data between testing markers and kinship. FarmCPU uses both the fixed-effect model and the random effect model iteratively. The $K$ derived from the markers is used to select the associated ones using the maximum likelihood method.

The population structure was inferred using a PCA using the population parameters previously determined (P3D) [31], while the relative kinship coefficients of individual genotypes based on identity (i.e., state) were estimated according to the VanRaden method [32] using all SNP markers.

The p-value obtained for each SNP was transformed into a logarithmic scale and later presented in circular Manhattan and quantile-quantile (QQ) plots created using CMplot packages

[33]. The significant associations between SNPs were corrected using the Bonferroni correction for multiple tests at the 5% and 1% levels of significance (0.05/23,078 and 0.01/23,078, respectively). Only traits that exceeded the threshold value and were consistent across GWAS methods have been reported in this study. The percent phenotypic variance (PV) explained by all significant SNPs was calculated as the squared correlation between the phenotype and genotype of the SNP.

Linkage disequilibrium was estimated using the correlation coefficients ($r^2$) between the pairs of loci on each chromosome using the *sommer* package in R [34], while the *IntAssoPlot* package [35] provided an integrated graphic of GWAS results with linkage disequilibrium (LD) and gene structure information. To investigate the LD decline, $r^2$ values were plotted as a function of genetic distance in kilobases (kb) using loess regression.

We used the *find.cluster* function of the adegenet package in R [36] to identify the number of clusters of the germplasm analyzed using the kinship matrix. This function implements successive K-means running with an increasing number of clusters (k) after transforming data using a PCA and then computing the goodness of fit (Bayesian Information Content) for each model to choose the optimal k. Then, the 3D PCA was used to group the cassava germplasm of each k cluster.

## 2.8 *In silico* SNP annotation

The candidate genes were searched within an 80-kb region (approximately 40 kb upstream and 40 kb downstream) of the detected significant SNP. The biological functions of genes/transcripts close to the significant SNPs were determined by searching the SNP sequences with proteins related to the starch pasting properties in the Phytozome version 12.1 database. The homology search was performed using BLASTx.

## 3 Results

### 3.1 Variation, correlation and heritability of the pasting properties

The 876 cassava accessions used in this study exhibited wide variation of all pasting properties (Table 1). The highest amplitudes were observed for Cold-PVisc (540.94 to 4,106.70 cP), with an average of 2,916.03 cP, and PeakVisc (3,101.59 to 6,701.79 cP), with an average of 4,809.67 cP. On the other hand, the amplitude values of SetBack (191.01 to 2,309.42 cP) and Hot-PVisc (351.41 to 2,692.87 cP) had moderately high magnitude. Additionally, PastTemp ranged from 63.82 to 75.47˚C, with an average of 70.05˚C, with low magnitude. Therefore, the analyzed germplasm generally consisted of a panel with wide phenotypic variation.

**Table 1. Minimum, maximum, and mean values of best linear unbiased predictors (BLUPs) and broad-sense heritability ($h^2$) for the pasting properties of cassava starch.**

| Trait | Minimum | Maximum | Mean ± SD | $h^2$ |
|---|---|---|---|---|
| PastTemp | 63.82 | 75.47 | 70.05 ± 1.28 | 0.66 |
| PeakVisc | 3101.59 | 6701.79 | 4809.67 ± 373.26 | 0.68 |
| Hot-PVisc | 351.41 | 2692.87 | 1759.11 ± 195.64 | 0.67 |
| BreDow | 1314.28 | 4767.70 | 3040.12 ± 297.92 | 0.69 |
| Cold-PVisc | 540.94 | 4106.55 | 2916.03 ± 345.24 | 0.76 |
| SetBack | 191.01 | 2309.42 | 1139.75 ± 229.12 | 0.71 |
| FinalVis | 532.66 | 3995.79 | 2868.85 ± 329.67 | 0.76 |

PastTemp: pasting temperature (˚C); PeakVisc: peak viscosity (cP); Hot-PVisc: hot-paste viscosity (cP); BreDow: breakdown viscosity (cP); Cold-PVisc: cool-paste viscosity at 50˚C (cP); SetBack: Setback (cP); FinalVis: final viscosity (cP).

In terms of broad-sense heritability ($h^2$), the variations for the numerous traits associated with pasting properties were of moderate to high magnitude (Table 1). The Cold-PVisc and FinalVis traits presented the highest $h^2$ (0.76), followed by SetBack (0.71). On the other hand, $h^2$ values of PastTemp, PeakVisc, Hot-PVisc, and BreDow were of a moderate magnitude (range: 0.66 to 0.69).

Although the PastTemp trait was negatively correlated with all other characteristics of cassava starch pasting properties, the correlation between PastTemp × Hot-PVisc was practically nil (Fig 1). The other pasting property traits had positive correlations with each other. We identified a high positive correlation between FinalVis × Cold-PVisc (r = 0.98). Other strong positive correlations (0.80< r <0.90) were identified between PeakVisc × BreDow, Hot-PVisc × (Cold-PVisc and FinalVis), and SetBack × (Cold-PVisc and FinalVis).

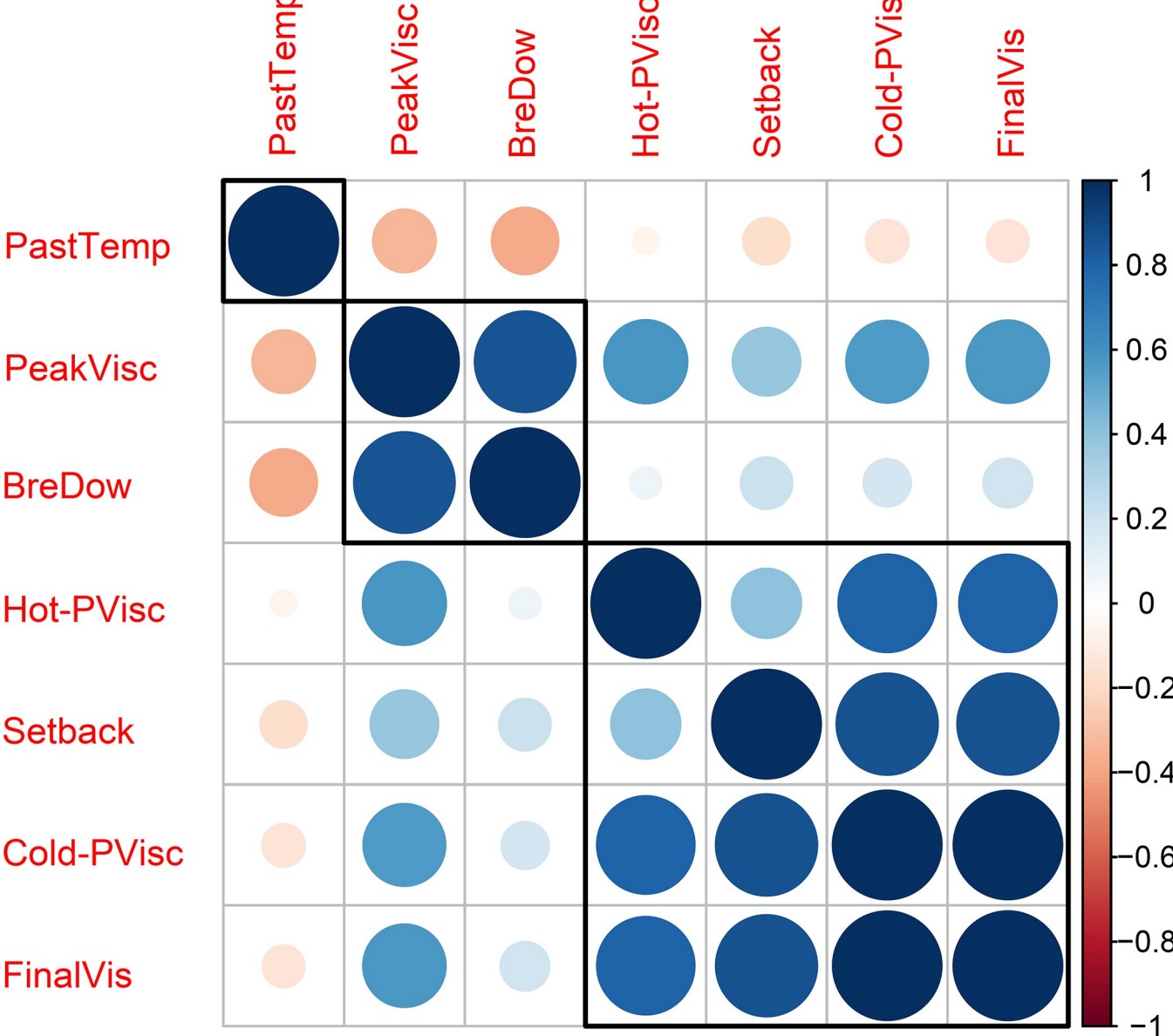

**Fig 1. Correlogram of the pasting property traits of cassava starch from 876 accessions.** Positive correlations are displayed as blue circles and negative correlations as orange circles. The sizes of the circles are proportional to the correlation coefficients.

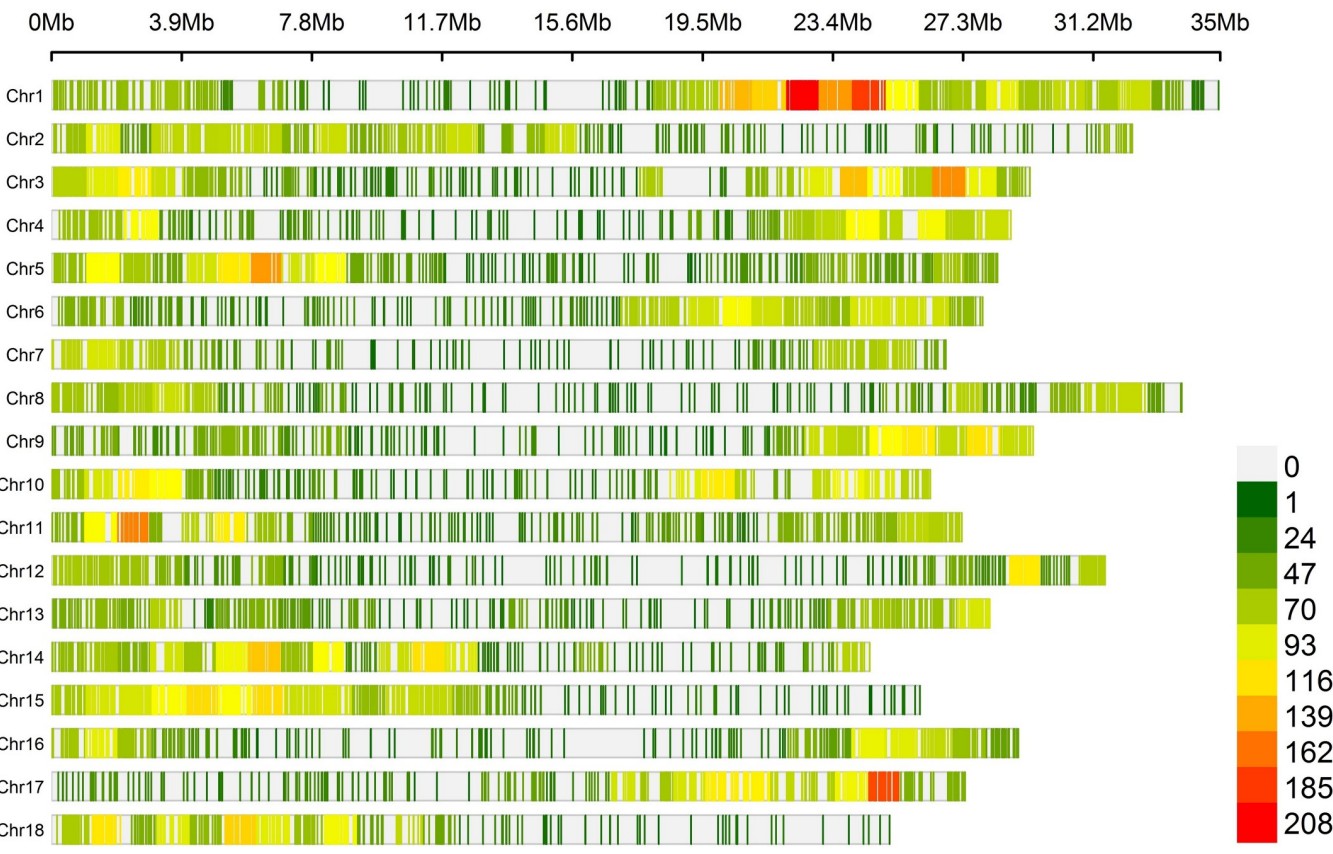

**Fig 2. SNP density plot across the 18 chromosomes of cassava representing the number of SNPs within a 1-Mb window size.** The horizontal axis shows the chromosome length (Mb), while the different colors depict SNP density.

The mean density of SNPs in the 18 chromosomes of *M. esculenta* was 1 SNP per 23.31 kb (Fig 2, S3 Table), where the highest and lowest marker densities were observed on chromosomes 1 (1 SNP per 16.39 kb) and 12 (1 SNP per 34.41 kb), respectively. On average, 1,282 SNPs were identified per chromosome, ranging from 882 (chromosome 7) to 2,133 SNPs (chromosome 1). SNPs were identified in similar numbers in all chromosomes, which improved the accuracy and reliability of the analyses.

The PCA plot based on the kinship matrix grouped the 876 cassava clones into five clusters (Fig 3, S1 Table). In general, weak relationships were identified among cassava accessions based on genomic kinship due to the evolutionary process and differentiated domestication processes of the accessions in this collection. The accessions were grouped into five clusters with sizes ranging from 115 to 271. Group 1, formed by 115 accessions, had the lowest number of individuals, of which 21% were improved varieties that largely originated from the Northeast, Midwest, and Southeast regions of Brazil. Group 1 also contained five clones from Colombia and one each from Nigeria and Venezuela, respectively. Group 2 was composed of 134 accessions, of which 87% were local varieties, and 12.37% were obtained from the germplasm collection (especially varieties from North and Northeast Brazil, where accessions from the Amazon regions represented 46% of the group). Group 3 had 151 accessions, of which 19% were improved varieties and 13, 2, 1, and 1 clones were from Colombia, Nigeria, Panama, and Venezuela, respectively. Group 4 was the largest, with 271 accessions, of which 82% were local varieties from the North, Northeast, and Southeast of Brazil. Finally, Group 5 had 205 accessions and the lowest proportion of improved varieties (13%)—most of these accessions were

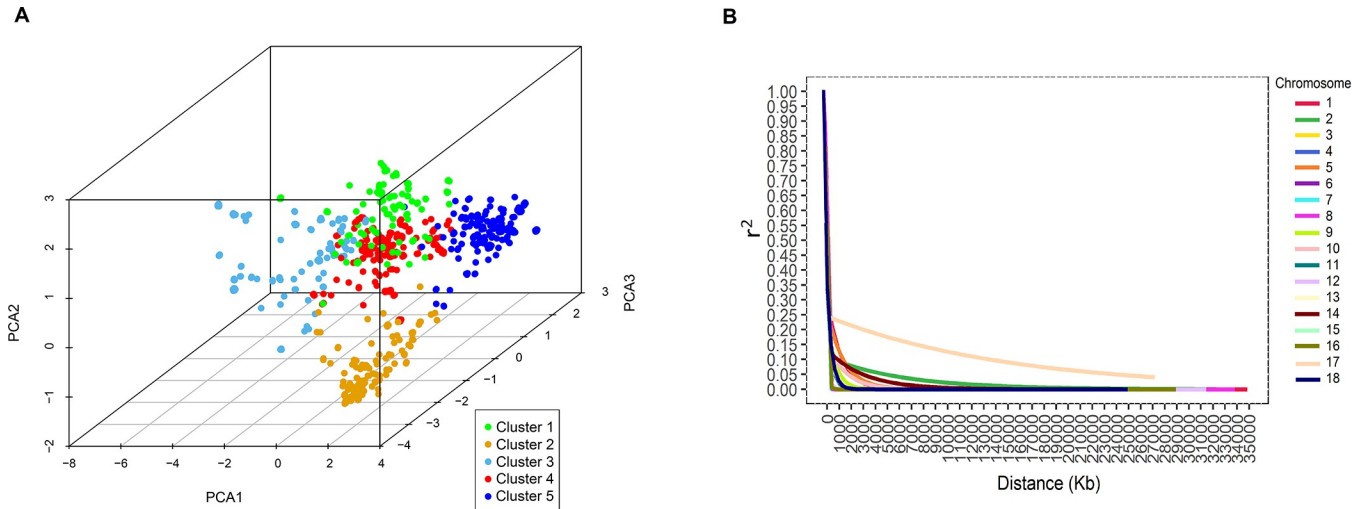

**Fig 3. Principal component analysis (PCA) and genome-wide linkage disequilibrium decay.** (A) 3D PCA of the kinship matrix obtained from 23,078 SNP markers of 876 cassava samples allocated in five clusters. (B) Genome-wide linkage disequilibrium decay of r2 values against physical distance (Mb). The lines in different colors indicate the adjustment of the loess regression to LD along the 18 cassava chromosomes.

collected from Northeast Brazil, while 11, 1, 1, and 1 each were from Colombia, Mexico, Nigeria, and Venezuela, respectively.

The LD values were calculated for each pair of SNPs (Fig 3). We observed that LD declined to background levels ($r^2 < 0.1$) of between 0.3 and 2.0 Mb in most cassava chromosomes (Fig 3). However, chromosome 17 showed atypical behavior, with an extensive megabase-scale LD. Therefore, the LD decayed to $r^2 < 0.1$ at a physical distance of ~20 Mb for chromosome 17. The relatively rapid LD decays observed in this germplasm panel, especially for chromosomes 4, 6, 7, 8, 9, 10, 11, 12, 13, 15, and 16, are indicative of its potential for reducing QTL intervals and fine mapping in several GWAS analyses.

### 3.2 Genome-wide association study for casava starch pasting properties

For the seven traits analyzed, 13 SNPs located on four different chromosomes (3, 8, 17, and 18) were found to be significantly associated with the pasting properties of cassava starch. Most traits (SetBack, FinalVis, PeakVisc, Cold-PVisc, and Hot-PVisc) had significant SNPs on chromosome 18, with the exception of PastTemp (Table 2). The results indicated that these traits are strongly correlated and possibly under the control of the same gene or even due to closely linked genes.

The regions of chromosome 18 covering the 2907312 to 3567816 positions had six SNPs significantly associated with cassava starch pasting properties (Table 2, Fig 4). SNP S18_3081635 was identified by the three GWAS methods for SetBack, FinalVis, Cold-PVisc, and Hot-PVisc (MLM method), thereby indicating that this SNP is capable of simultaneously influencing the expression of these four traits, possibly due to the high correlation between these characteristics. SNP S18_3081635 had a positive effect on the expression of all of these traits (i.e., 244.10, 406.92, 416.01, and 180.27 cP for SetBack, FinalVis, Cold-PVisc, and Hot-PVisc, respectively).

SNPs S18_3399799 and S18_3399801 (separated by only two bases) also showed significant and negative effects for SetBack (-250.99 cP), FinalVis (-439.70 cP), Cold-PVisc (-454.54 cP), and Hot-PVisc (-215.79 cP). Other SNPs with high LD were S18_3567791 and S18_3567816, which were separated by only 25 bases. Similarly, the latter SNPs were significantly associated

**Table 2. Single-nucleotide polymorphism (SNP) markers associated with cassava starch pasting properties based on the MLM, MLMM, and FarmCPU methods.**

| Trait | SNP | Chromosome | Position | Alleles | GWAS method | | | | | |
|---|---|---|---|---|---|---|---|---|---|---|
| | | | | | MLM | | | MLMM | FarmCPU | |
| | | | | | -log10(p-value) | r2 | Effect | -log10(p-value) | -log10(p-value) | Effect |
| Setback | S18_2907312 | 18 | 2907312 | A/C | 6.62 | 0.13 | 246.76 | | | |
| Setback | S18_3081635 | 18 | 3081635 | G/A | 11.31 | 0.15 | 244.10 | 13.69 | 9.60 | 185.86 |
| Setback | S18_3399799 | 18 | 3399799 | G/T | 9.22 | 0.14 | 250.99 | | | |
| Setback | S18_3399801 | 18 | 3399801 | A/C | 9.22 | 0.14 | 250.99 | | | |
| Setback | S18_3567791 | 18 | 3567791 | G/C | 6.08 | 0.13 | 297.38 | | | |
| Setback | S18_3567816 | 18 | 3567816 | T/A | 8.00 | 0.14 | 168.10 | | | |
| Setback | S18_17771889 | 18 | 17771889 | C/T | | | | | 5.95 | 81.72 |
| PastTemp | S3_5856578 | 3 | 5856578 | G/A | | | | | 6.93 | 0.46 |
| PastTemp | S8_32339820 | 8 | 32339820 | A/T | | | | | 6.07 | 1.26 |
| FinalVis | S17_17327003 | 17 | 17327003 | G/C | | | | 6.77 | | |
| FinalVis | S18_2907312 | 18 | 2907312 | A/C | 8.12 | 0.16 | 437.50 | | | |
| FinalVis | S18_3081635 | 18 | 3081635 | G/A | 12.58 | 0.18 | 406.92 | 16.28 | 6.76 | 249.76 |
| FinalVis | S18_3399799 | 18 | 3399799 | G/T | 11.33 | 0.17 | 439.70 | | | |
| FinalVis | S18_3399801 | 18 | 3399801 | A/C | 11.33 | 0.17 | 439.70 | | | |
| FinalVis | S18_3407893 | 18 | 3407893 | T/C | 7.11 | 0.15 | 335.88 | | | |
| FinalVis | S18_3408138 | 18 | 3408138 | C/A | 6.67 | 0.15 | 479.83 | | | |
| FinalVis | S18_3567791 | 18 | 3567791 | G/C | 8.19 | 0.16 | 541.96 | | | |
| FinalVis | S18_3567816 | 18 | 3567816 | T/A | 10.60 | 0.17 | 306.48 | | | |
| PeakVisc | S18_13522313 | 18 | 13522313 | G/A | 5.90 | 0.06 | 230.44 | 6.93 | 8.61 | 254.66 |
| Cold-PVisc | S17_17327003 | 17 | 17327003 | G/C | | | | 7.56 | | |
| Cold-PVisc | S18_2907312 | 18 | 2907312 | A/C | 7.83 | 0.15 | 448.79 | | | |
| Cold-PVisc | S18_3081635 | 18 | 3081635 | G/A | 12.04 | 0.17 | 416.01 | 15.43 | 9.77 | 291.19 |
| Cold-PVisc | S18_3399799 | 18 | 3399799 | G/T | 11.06 | 0.17 | 454.54 | | | |
| Cold-PVisc | S18_3399801 | 18 | 3399801 | A/C | 11.06 | 0.17 | 454.54 | | | |
| Cold-PVisc | S18_3407893 | 18 | 3407893 | T/C | 6.92 | 0.15 | 346.62 | | | |
| Cold-PVisc | S18_3408138 | 18 | 3408138 | C/A | 6.37 | 0.15 | 489.69 | | | |
| Cold-PVisc | S18_3567791 | 18 | 3567791 | G/C | 7.68 | 0.15 | 548.06 | | | |
| Cold-PVisc | S18_3567816 | 18 | 3567816 | T/A | 10.34 | 0.17 | 316.65 | | | |
| Hot-PVisc | S18_2907312 | 18 | 2907312 | A/C | 5.78 | 0.07 | 207.30 | | | |
| Hot-PVisc | S18_3081635 | 18 | 3081635 | G/A | 7.81 | 0.09 | 180.27 | | | |
| Hot-PVisc | S18_3399799 | 18 | 3399799 | G/T | 8.40 | 0.09 | 215.79 | 9.71 | | |
| Hot-PVisc | S18_3399801 | 18 | 3399801 | A/C | 8.40 | 0.09 | 215.79 | | | |
| Hot-PVisc | S18_3407893 | 18 | 3407893 | T/C | 5.86 | 0.08 | 175.23 | | | |
| Hot-PVisc | S18_3567791 | 18 | 3567791 | G/C | 6.00 | 0.08 | 266.68 | | | |
| Hot-PVisc | S18_3567816 | 18 | 3567816 | T/A | 7.84 | 0.09 | 150.41 | | | |

MLM (mixed linear model), MLMM (multi-locus mixed-model), and FarmCPU (Fixed and random model for circulating probability unification).

with SetBack, FinalVis, Cold-PVisc, and Hot-PVisc, with a positive effect for all of these traits. Although separated by approximately 660 kb, SNP S18_2907312 was also associated with Set-Back, FinalVis, Cold-PVisc, and Hot-PVisc, but with a negative effect for all of these traits (i.e., -246.76, -437-50, -448.79, and -207.30 cP, respectively) (Table 2). Notably, these SNPs, except S18_3399799, were only identified by the MLM method. On the other hand, according to the MLMM method, SNP S17_17327003 was significantly associated with positive effects for the FinalVis and Cold-PVisc traits (Table 2, Fig 4).

The FarmCPU method also revealed the presence of another SNP that was positively associated (81.72 cP) with SetBack (Table 2, Fig 4) as well as two SNPs on chromosomes 3 (S3_5856578) and 8 (S8_32339820) with negative (-0.46°C) and positive (1.26°C) effects on

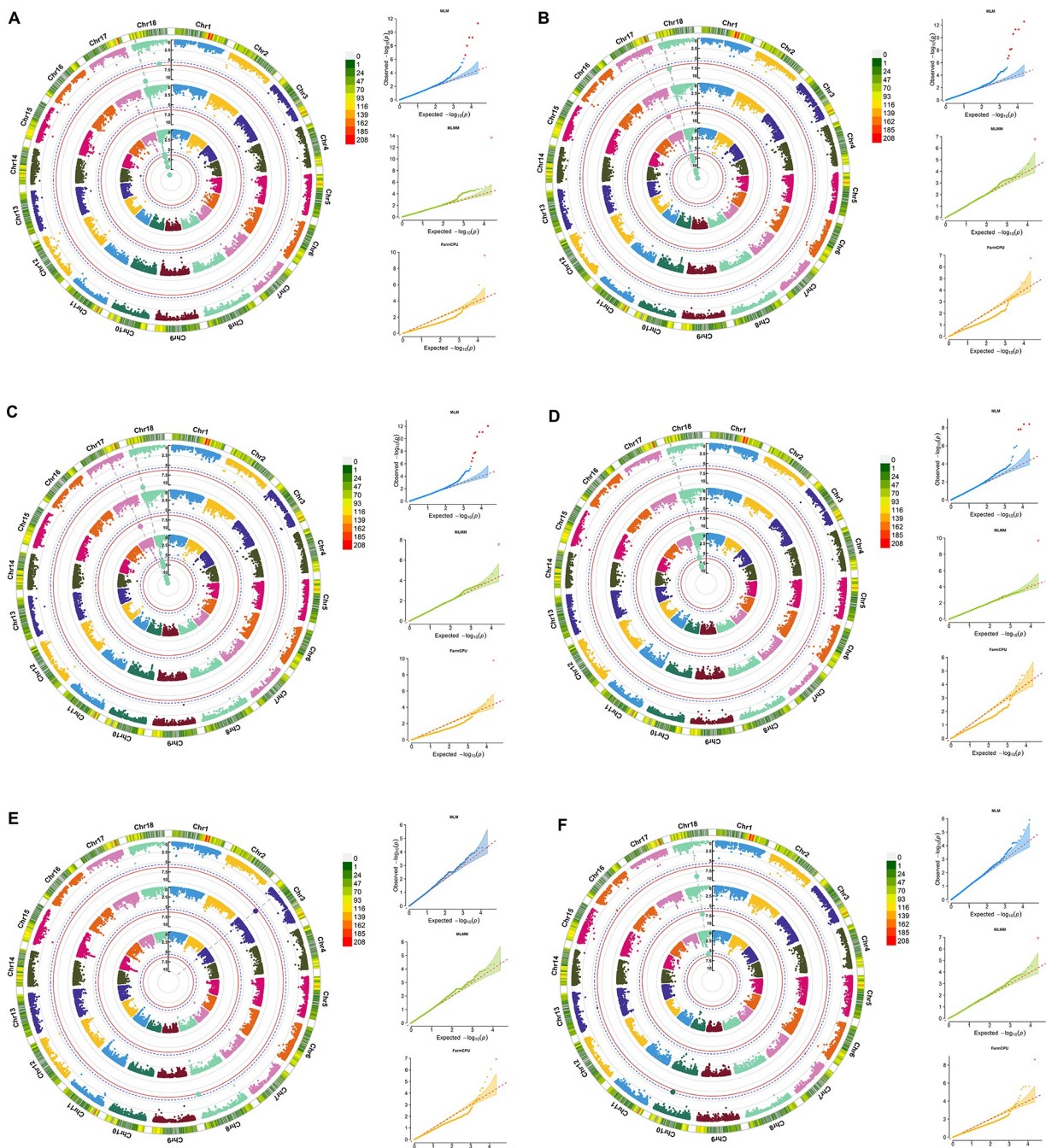

**Fig 4.** Circular Manhattan plot indicating the single-nucleotide polymorphisms (SNPs) associated with (A) setback, (B) final viscosity, (C) cool paste viscosity, (D) hot paste viscosity, (E) pasting temperature, and (F) peak viscosity, by analyzing the pasting properties of starch from a set of 876 cassava accessions. The locations of SNPs on the chromosome and the association test (-log₁₀(p)) are plotted on the x-axis and y-axis, respectively. The blue dotted line and red line indicates the Bonferroni correction at $p < 0.05$ and $p < 0.01$, respectively. The chart (right) presents the quantile-quantile (QQ) plot of the observed and expected $p$-values from association analyses based on the mixed linear model (MLM), multi-locus mixed model (MLMM), and fixed and random effects model for circulating probability unification (FarmCPU) methods.

PastTemp, respectively. On the other hand, only one SNP significant for the PeakVisc trait was found, on chromosome 18 (S18_13522313). S18_13522313 was consistently identified by the three GWAS methods with negative effects (-254.66 cP).

Even with the high correlation of PeakVisc × BreDow (0.86) (Fig 1), no SNPs associated with BreDow were identified. This may indicate that this correlation is due to pleiotropic effects and not physical linkage.

The variance explained by these markers was very close for SetBack, FinalVis, and Cold-PVisc ($r^2$ range: 0.13 to 0.18), while the low variances of the PeakVisc and Hot-PVisc traits were explained by the SNPs ($r^2$ range: 0.06 to 0.09) (Table 2).

A more detailed analysis of chromosome 18 indicated a block of markers in high LD between positions 2907312 and 3567816. The regional marker-trait associations of SetBack (S1 Fig), FinalVis (S3 Fig), Cold-PVisc (S5 Fig), and Hot-PVisc (S6 Fig) demonstrated similar behavior regarding the LD blocks of the SNP markers.

In addition to this genomic block associated with several pasting property traits, another portion of chromosome 18 was also identified, which was specifically associated with SetBack at position 17771889 using the FarmCPU method (S1 Fig). For PastTemp, both SNPs S3_5856578 and S8_32339820 were present in regions with relatively low genomic coverage, and there were no other SNPs with high LD among them (S2 Fig). Another SNP located in a region of low marker density was SNP S18_13522313, which was identified on chromosome 18 for PeakVisc using all three GWAS methods (S4 Fig).

The genotypes of SNP S17_17327003 exhibited significant differences for Cold-PVisc and FinalVis (i.e., the CG contents were approximately 379 and 329 cP higher than in the GG genotypes) (Fig 5). On the other hand, SNPs S18_2907312, S18_3081635, S18_3399799, S18_3399801, S18_3567791, and S18_3567816 showed that homozygous genotypes of favorable alleles (A, G, G, A, G, and T, respectively) had high Cold-PVisc, FinalVis, Hot-PVisc, and SetBack, while heterozygous and homozygous alternative alleles exhibited medium and low pasting property traits, respectively. The same trend was identified for SNP S18_3408138, in which the AA, AC, and CC genotypes had ~1,163 cP, 2,315 cP, and 2,937 cP, respectively, for Cold-PVisc, and 1,141 cP, 2,285 cP, and 2,890 cP, respectively, for FinalVis. These results indicated that the genetic control of Cold-PVisc, FinalVis, Hot-PVisc, and SetBack generally had additive effects. Therefore, increasing the frequency of those favorable alleles in the right direction would significantly improve the quality of cassava starch.

In contrast, only the homozygous form of the favorable allele (T) of SNP S18_3407893 was significantly different from the other genotypes for Cold-PVisc (CT = 1,560 cP and CC = 1,318 cP), FinalVis (CT = 2,441 cP and CC = 1,962 cP), and Hot-PVisc (CT = 2,416 cP and CC = 1,930 cP). The same behavior for PastTemp was identified for SNP S3_5856578, in which the homozygous form of the favorable allele (A) was significantly different from other genotypes (AG = 70.09°C and GG = 69.97°C). The other significant SNP for PastTemp (S8_32339820) had only two genotypes, with ~1.2°C difference between AA and AT. On the other hand, for SNP S18_17771889, the genotypes TT and CT were similar for SetBack (1,240 cP and 1184 cP, respectively), while the alternative homozygous form was significantly different from the other genotypes (1,126 cP). The same was identified for the significant SNP for PeakVisc (S18_13522313). The genotypes AA and AG of S18_13522313 had ~5,000 cP, while the genotype GG had ~4,700 cP. Therefore, it is possible to speculate that the latter SNP on chromosome 18, which is completely separated from the other LD block, had a dominance effect on the expression of the setback trait in cassava starch.

### 3.3 *In silico* annotation of SNPs associated with pasting properties of cassava starch

For PastTemp, we identified 4 and 12 transcripts, in SNPs S3_5856578 and S8_32339820, respectively, most of which have unknown functions (S4 Table). The transcript of SNP

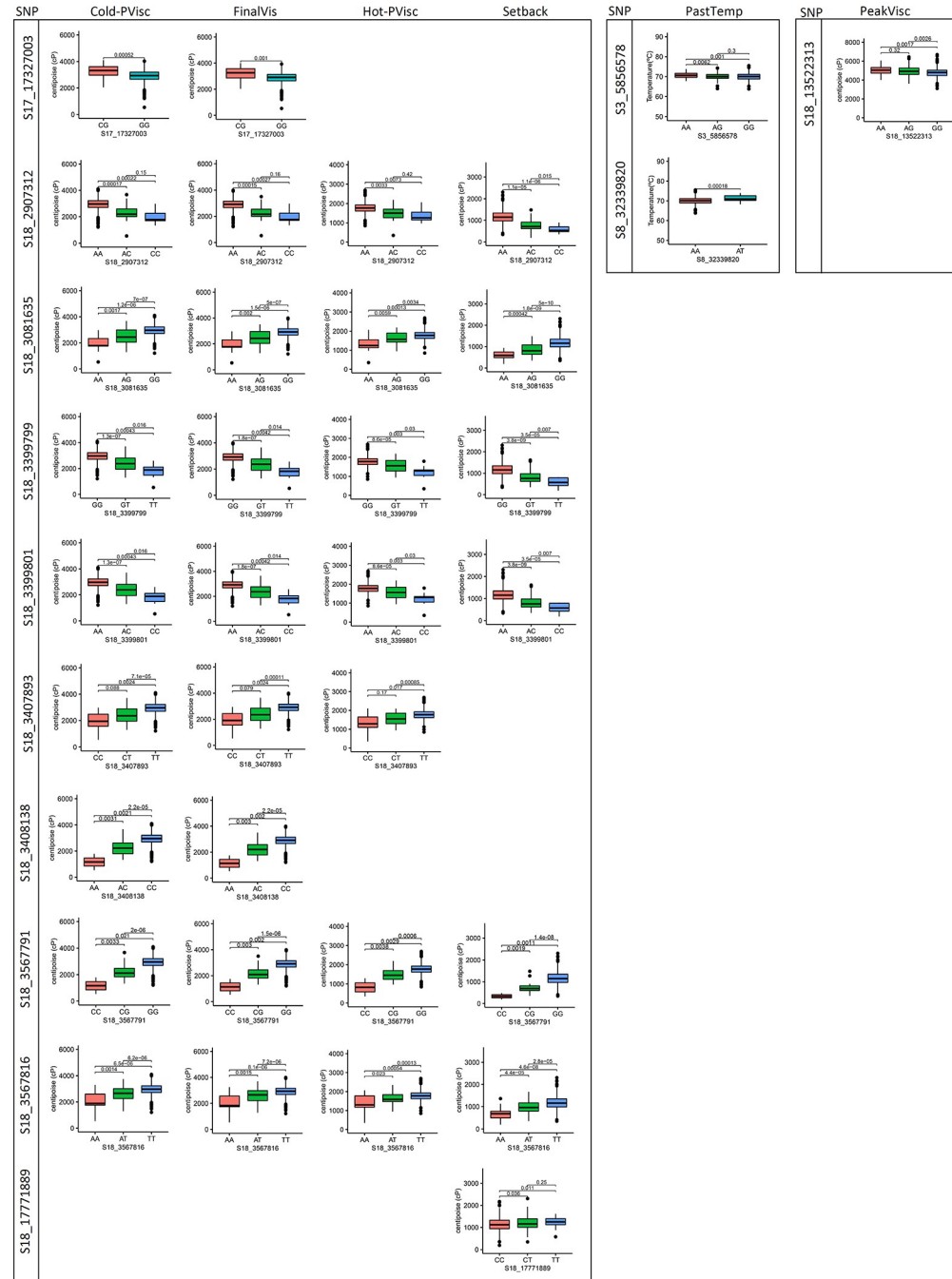

**Fig 5. Analysis of build-up effect for different SNP genotypes associated with pasting temperature (PastTemp), peak viscosity (PeakVisc), hot paste viscosity (Hot-PVisc), cool paste viscosity (Cold-PVisc), final viscosity (FinalVis), and setback (Setback).** The x-axis represents the SNP genotypes carried by the cassava accessions, and the y-axis represents trait mean value. The p-value of the t-test comparison (p<0.05) is plotted for each comparison.

S3_5856578 is associated with polyketide cyclase/dehydrase and lipid transport (Manes.03G059600), which is responsible for catalyzing the hydrolysis of various bonds (e.g., c-o, c-n, c-c) and the regulatory component of the ABA receptor. The other transcript of SNPs S3_5856578 (Manes.03G059800) is associated with cotton fiber expressed protein, which belongs to a family of several plant proteins of unknown function. In SNP S8_32339820, only

4 out of 12 transcripts have known functions: Manes.08G161600 is related to the 26s protease regulatory subunit; Manes.08G161800 and Manes.08G161900 are related to 3'-5' exonuclease; and Manes.08G162100 is related to phosphatidylinositol 3- and 4-kinase.

SNPs S17_17327003 and S18_3408138 were significantly associated with Cold-PVisc and FinalVis and had 10 and 1 transcripts, respectively (S4 Table). Most transcripts from S17_17327003 have known functions that are largely related to organic solute transporters (Manes.17G040100), beta-galactosidase and glycosyl hydrolase family 35 (GH35) (associated with starch and sucrose metabolism) (Manes.17G040300), s1/p1 nuclease (Manes.17G040400), secretory carriers (Manes.17G040700), a protein of unknown function (Manes.17G040800), the DNA-binding domain (Manes.17G040900), and repeat-containing subunit B1 homolog (Manes.17G041000). Additionally, Manes.18G039200 from SNP S18_3408138 is related to proprotein convertase subtilisin/kexin.

The traits Hot-PVisc, Cold-PVisc, SetBack, and FinalVis exhibited a LD block with several associated SNPs. Therefore, a large number of transcripts are correlated with these traits for cassava starch pasting properties. SNPs S18_3399799, S18_3407893, and S18_3567791 exhibited only one transcript associated with these four traits. The function of Manes.18G039300 relates to the subtilase family and peptidase inhibition, while Manes.18G039100 relates to ergosterol biosynthetic protein 28-related functions, and Manes.18G040700 relates to an unnamed family. Five out of six transcripts around SNP S18_3399801 are associated with the protein kinase domain (Manes.18G039400), helix-loop-helix DNA-binding domain (Manes.18G039500 and Manes.18G039600), and RING finger domain (Manes.18G039700 and Manes.18G039900).

Furthermore, seven of the nine transcripts of S18_3567816 have functions associated with protein tyrosine kinase and leucine-rich repeat (Manes.18G040800), WRKY DNA-binding and the plant zinc cluster domain (Manes.18G041000), cysteine protease family C1-related and a component of protease-inhibitor complex-related (Manes.18G041100), mannose-p-dolichol utilization defect 1 protein and lec35-related (MPDU1) (involved in general metabolism and glycosyltransferase activities) (Manes.18G041200), the F-box domain (Manes.18G041400), and the cyclin, c- and n-terminal domains (Manes.18G041600).

In the case of SNP S18_3081635, all 11 transcripts have known functions and are associated with the transmembrane amino acid transporter protein (Manes.18G035300), Dof domain and zinc finger (Manes.18G035400), protein phosphatase 2c (Manes.18G035500), BRI1 kinase I (Manes.18G035600), protein LPIN-1 (Manes.18G035700), and EF-hand calcium-binding domain-containing protein (Manes.18G035800, Manes.18G035900, Manes.18G036000, Manes.18G036100, Manes.18G036200, and Manes. 18G036300). Of the 15 transcripts on SNP S18_2907312, six have an unknown function, and three are related to patatin-like phospholipase (Manes.18G033200, Manes.18G033300, and Manes.18G033400). The remaining transcripts are associated with NAD-dependent epimerase/dehydratase (Manes.18G032300), a domain of unknown function (Manes.18G032800), a member of the 'GDXG' family of lipolytic enzymes, and carboxylesterase 2-related (Manes.18G032900 and Manes.18G033000), the 26s proteasome non-ATPase regulatory subunit 14 and translation initiation factor 3-related (Manes.18G033100), and heme oxygenase (Manes.18G033500). Additionally, there were no known transcripts associated with SNP S18_17771889 for the setback trait.

## 4 Discussion

### 4.1 Phenotypic variation and heritability of pasting properties of cassava starch

The differences observed in the pasting properties of cassava starch provide an important opportunity for the selection of accessions that can be used for various industrial applications.

Identifying the existence of groups of accessions with different starch properties is an important step for cassava breeding programs, since these groups indicate the most promising accessions for use in crosses to develop new varieties with the desired traits. On the other hand, the gains expected from this selection process are dependent on the heritability of the traits.

The moderate values of $h^2$ for PastTemp, PeakVisc, Hot-Pvisc, and BreDow (0.66, 0.68, 0.67, and 0.69, respectively) and the high values of Cold-PVisc, FinalVis (0.76), and SetBack (0.71) indicate that a large part of the observed phenotypic variance is due to genetic factors. Therefore, there is potential to transfer these traits to improved varieties with relative ease.

For the pasting properties of corn starch (*Zea mays* L.), [37] reported moderate $h^2$ values of PastTemp, BreDow, and FinalVis (0.54, 0.56, and 0.61, respectively) and high $h^2$ values of PeakVisc (0.70), SetBack (0.72), and Hot-PVisc (0.77). On the other hand, [38] reported high $h^2$ values for starch pasting properties in rice (*Oryza sativa* L.), varying from 0.84 (PastTemp) to 0.92 (BreDow). In cassava, other authors have reported similar $h^2$ values to those found in the present study for PastTemp, PeakVisc, Hot-Pvisc, and BreDow (0.77, 0.49, 0.45, and 0.51, respectively) [11]. In the case of PastTemp, they did not identify a genotype × environment interaction.

Indeed, starch pasting properties can be affected by environmental conditions since the contents of amylose and amylopectin (the main constituents of starch [9]) vary according to the environment [39]. In the present study, the environment had little influence on the expression of starch properties in the cassava germplasm, predominantly from Latin America, possibly because the accessions were cultivated in only one region, which may have been reflected in the high heritability of these traits.

Regarding the correlations between the traits associated with starch pasting properties, other studies of cassava have also shown a negative correlation between PastTemp × PeakVisc and Hot-PVisc (r = -0.06 and -0.19, respectively) when analyzing biofortified cassava accessions [40]. In another study, [41] also identified a negative correlation between PastTemp × FinalVisc and BreDow (r = -0.58 and -0.55, respectively). In addition, the latter authors reported a strong and positive correlation between PeakVisc × BreDow (r = 0.98) and a high positive correlation between Setback × BreDow (r = 0.64). According to [42], in general the high correlations among starch pasting properties are due to the nature of the RVA analysis. A potential explanation for the high positive correlation among some of those traits may be that when the temperature reaches 95°C and decreases to 50°C (Cold-PVisc), the molecules tend to reorganize (SetBack), which can increase viscosity and maintain it until the end of the analysis (FinalVis).

## 4.2 Density of markers, population structure and linkage disequilibrium

The high density of markers present in the cassava genome significantly affects GWAS efficiency, since in general it tends to improve the resolution of the mapping, and thus the accuracy of genomic associations [7, 43]. This effect of marker density is determined by the extent of linkage disequilibrium (LD), since low LD requires a higher density of markers to provide better mapping resolution. However, when LD is higher, a lower density of markers can be used, although, in some cases this leads to low resolution [44]. The mean marker density observed in this study, even after the quality filters were adopted, showed broad representativeness of the cassava genome, with a distribution of 1 SNP/23.31 kb.

Linkage disequilibrium (LD) studies have been used routinely with cassava as a complementary approach to studies of genomic selection [45], genetic diversity and population structure [46, 47], QTL mapping [13], as well as GWAS studies [15, 17, 18, 23, 47, 48]. The extent of LD is a determining factor for GWAS efficiency, and among the methods used to visualize LD decay, the coefficient of determination ($r^2$) approach has been widely used [49]. This method

is based on the plotting of LD estimates as a function of the genetic distance in base pairs (bp) between SNPs [50, 51].

In the present study, LD decay ($r^2 < 0.1$) occurred in the region between 0.3 and 2.0 Mb, although the LD decay from chromosome 17 to $r^2 < 0.1$ occurred at a physical distance of ~ 20 Mb. By analyzing 3354 cassava landraces and modern breeding lines from the Embrapa Cassava Germplasm Bank using 27,045 SNPs, [24] also reported that LD dropped to a background level at around 0.3 Mb across the genome and that a higher chromosomal LD extent was observed on chromosomes 5 and 17, with the slowest decay. These authors also speculated about the introgression of *M. glaziovii* in Brazilian germplasm collections on chromosome 17. Similar behavior was observed by [52] when analyzing other clonally propagated crops (potato). The latter work comprised a germplasm panel with 351 diverse potato genotypes, in which the chromosomes exhibited modest LD across the whole chromosomal region, ranging from 2.54 Mb (chromosome 9) to 4.68 Mb (chromosome 3), whereas chromosome 8 exhibited a conserved LD of 20.04 Mb.

In another study of cassava, [15] used a panel of 672 accessions from Africa genotyped with 72,279 SNP loci and reported that the LD extended to a distance of 2Mb ($r^2 < 0.10$). Therefore, our results demonstrate a lesser extent of LD in the cassava germplasm from Latin America, which can be quite useful for trait introgression. Throughout the genome, the extent of LD can be highly variable both within and between populations of the same species. These differences are related, in particular, to the complexity and size of the genome as well as the number of markers used to achieve all existing genetic variations [53]. Evolutionary factors such as mutation, recombination, selection, genetic drift, and crossing patterns also have a strong impact on the extent of LD [54, 55]. In general, prior understanding of the LD decay pattern improves statistical power and decreases the rate of false positives in genomic association analyses that target the discovery of QTL-bound genes [56].

## 4.3 Genome-wide association study for cassava starch traits

Based on the analysis of this germplasm, the 13 SNPs associated with the PeakVisc, Hot-PVisc, SetBack, Cold-PVisc, PastTemp, and FinalVis traits can guide MAS to improve starch pasting properties. To the best of our knowledge, this is one of the first reports of the use of GWAS for mapping these traits in cassava starch. Of the 13 SNPs, 10 are on chromosome 18, and the rest on chromosomes 3, 8, and 17. Of the 10 SNPs on chromosome 18, eight are in an LD block that covers ~660 kb from SNP S18_2907312 to S18_3567816. In the modern cassava breeding pipeline, the MAS of these co-associated genetic loci could simultaneously improve the multi-target traits of cassava root quality. All told, six different genomic regions in high LD with the PeakVisc, Hot-PVisc, SetBack, Cold-PVisc, PastTemp, and FinalVis traits in cassava starch were identified. Similarly, in waxy maize, there are also reports of the presence of QTL clusters co-associated with two or more RVA parameters in conventional biparental QTL mapping [57]. In total, eight QTLs, for BreDow, PeakVisc, SetBack, FinalVis, SetBack, pasting time, and trough viscosity were co-located on chromosomes 2, 3, 4, 5, 7, and 9.

The first attempt to identify QTLs and candidate genes associated with cassava starch paste properties was performed by [11], using a traditional biparental population with 100 lines. In that study, the genetic mapping based on the mean of the field trials identified 15 QTLs affecting five starch pasting viscosities. The most important QTL was co-localized in linkage group 1 for PastTemp (qPT.1LG1). Although most of the reported QTLs are present in some environments only, [11] also reported one QTL controlling several of the starch pasting properties, such as PeakVisc, Hot-PVisc, Cold-PVisc, and SetBack (RY08.1LG1).

The $r^2$ explained by the SNPs related to cassava starch paste properties ranged from 0.13 to 0.18 for SetBack, FinalVis, and Cold-PVisc, and from 0.06 to 0.09 for PeakVisc and Hot-PVisc

(Table 2). A similar range of $r^2$ explained by molecular markers was found in maize, in which 72 QTLs detected for seven RVA parameters accounted for 0.06 to 0.18 of the phenotypic variation by analyzing 198 families derived from a cross between two waxy maize parents [57]. In contrast, in biparental QTL mapping of cassava, [11] reported a large phenotypic variance explained by the qPT.1LG1 (0.48) for PastTemp and a range of 0.13 to 0.37 for PeakVisc, Hot-PVisc, Cold-PVisc, and SetBack. However, the limited sample size in mapping populations has adverse consequences on QTL inferences. Through simulations, [58] demonstrated that a limited sample size leads to serious underestimation of the number of QTLs that affect the trait, and secondarily, to overestimation of the effect of any QTL. This could explain the high phenotypic variance of the cassava starch paste properties in the study by [11], who used only 100 lines of an F1 mapping population (Huay Bong 60 × Hanatee).

Regarding the gene action, it is possible to speculate that the traits Cold-PVisc, FinalVis, Hot-PVisc, and SetBack have additive effects due to the significant differences between the genotypes that have one or two copies of the favorable alleles, especially in the LD block between SNP S18_2907312 and S18_3567816. On the other hand, genotypes with at least one copy of the alternative alleles, e.g., TT and CT for S18_17771889 and AA and AG for S18_13522313, showed higher SetBack and PeakVisc averages, respectively (without a significant difference between these two genotypes), whereas the homozygous genotypes for the favorable alleles showed the lowest averages for SetBack (CC) and PeakVisc (GG). Therefore, a dominance effect may be responsible for the genetic action of these two most distant SNPs on chromosome 18.

The traits Hot-PVisc, Cold-Pvisc, and FinalVis are indicative of the ability of the sample to form a viscous paste after the cooking and cooling process. In general, higher Cold-PVisc and FinalVis occur due to the reassociation of amylose molecules during the cooling phase [5]. Also, they are related to the final texture of the product, determining food quality and acceptability, especially for foods that need to be stored for long periods [59]. In the present study, the LD block from SNP S18_2907312 to S18_3567816 had a high additive effect on these traits; this might support early selection of genotypes to improve these attributes.

Generally, cassava starches with lower PastTemp have some advantages for industrial food and non-food processes, since they form pastes more easily, reducing the energy cost of starch processing for production of certain foods [60]. Therefore, cassava accessions with a sufficient variation in PastTemp should be able to provide a wider range of options for starch industries. Therefore, SNPs on chromosomes 3 (S3_5856578) and 8 (S8_32339820), with negative (-0.46°C) and positive (1.26°C) effects, respectively, for PastTemp can be useful genomic regions underlying PastTemp for MAS.

On the other hand, the SetBack trait should also increase under this selection, so there will be a greater retrogradation tendency. Although this feature is not desired in some industrial applications, especially producing foods that cannot lose water during storage, starch with high values of SetBack can be used to increase the dietary fiber content as well as modulate the glycemic index due to the lower digestibility of the starch, especially in diet foods [5, 61].

For PeakVisc, the dominant effect of the A allele of SNP S18_13522313 can support the selection of genotypes with high-viscosity starches for the development of products with a high thickening power [62]; for example, puddings and gumdrops, as well as fat substitutes in the manufacture of ice creams [63, 64]. On the other hand, accessions with the G allele, which has a negative effect, can be used for the manufacture of products that require low viscosity, such as cheese-like products, jellies, processed meats, and extruded snacks, since they generally present lower granule swelling, higher solubility, a higher gelatinization temperature range, and a lower retrogradation tendency [65].

Starch pasting properties are characteristics predominantly controlled by a few genes with a significant effect on phenotypic expression [11]. Previous studies have reported that

pleiotropic effects and QTL hotspots are the primary factors affecting pasting properties in rice [39]. The pleiotropic effects are important for the genetic improvement of crops, especially when the gene affects multiple traits of interest, since they allow effective direct selection of genotypes [38]. However, in cassava the moderate to high heritability found in the present study and the large LD block on chromosome 18, and a few other regions associated with pasting traits, suggest that a set of overlapping QTLs can control the same traits. Therefore, it is possible that these correlated effects are primarily due to the genetic linkage instead of pleiotropy. Fine-mapping is a strategy that can be used in future studies to determine the genetic variant responsible for pasting traits.

In the present study, even with the high correlation between PeakVisc × BreDow (r = 0.86), no SNPs were associated with BreDow. In rice, nine QTLs associated with BreDow were detected in different chromosomes and environments, and QTL qBDV11 explained the highest proportion of phenotype variation (43.88%) [39]. Therefore, environmental and even pleiotropic effects may be responsible for the expression of BreDow in cassava starch. However, future studies including cassava starch obtained in different environments × years of cultivation, in addition to another set of germplasm and high-density SNPs, should be analyzed to confirm this hypothesis.

## 4.4 Comparing different GWAS methods

The MLM method detected the largest number of SNPs significantly associated with Cold-PVisc, FinalVis, Hot-PVisc, PeakVisc, and/or Setback on chromosome 18 (S18_2907312, S18_3081635, S18_3399799, S18_3399801, S18_3407893, S18_3408138, S18_3567791, S18_3567816, and S18_13522313), while the MLMM method identified only four SNPs, on chromosomes 17 and 18 (S18_3081635, S18_3399799, S18_13522313, and S17_17327003), associated with Cold-PVisc, FinalVis, Hot-PVisc, PeakVisc, and/or Setback. Finally, the Farm-CPU method identified five SNPs, associated with Cold-PVisc, FinalVis, PastTemp, PeakVisc, and/or Setback, on chromosomes 3, 8, and 18. Other studies have demonstrated the need to evaluate different GWAS methods to detect genomic regions associated with certain traits. [66] identified candidate QTLs for ionomic and agronomic traits in accessions of the USDA rice mini-core collection using univariate (GLM and MLM) and multivariate (MLMM and FarmCPU) methods. These authors demonstrated that each of the four GWAS methods identified unique and common QTLs, suggesting that different approaches should be used to complement the identification of QTLs in complex traits.

In general, the MLM, MLMM, and FarmCPU methods have different approaches to detect genomic regions associated with certain traits. The MLM method integrates the population structure and incorporates hidden relationships in the model. The MLM method adds the genetic effect of individuals as a random cofactor with the variance structure defined by the K among individuals, and no cofactors are adjusted by the marker tests [67]. The MLM method tests multiple markers simultaneously by fitting pseudo QTNs in addition to the testing markers in a stepwise MLM. Then, genetic markers can be tested one at a time with all pseudo QTNs included as covariates [68]. On the other hand, the FarmCPU method divides the MLM into a fixed-effect (FEM) and random effect (REM) model, using them iteratively to increase the detection power of candidate genes associated with the structure of the population. There are several reports of the greater statistical power of FarmCPU compared to the GLM and MLM methods to evaluate populations with contrasting population structures [69].

Although there have been important improvements in the statistical power of GWAS methods under certain assumptions, the choice of the most appropriate methods for a given species and dataset is still a task that must be investigated. In the specific case of cassava starch paste

properties, although we selected the three methods with the highest statistical power for detecting QTLs, there was a large discrepancy in the significance of most SNPs. Therefore, it will still be necessary to validate the genomic regions we found through gene expression analysis or even the development of KASP markers to complement the statistical and biological evidence for the practical use of these markers in molecular-assisted selection processes.

## 4.5 Transcripts most likely associated with cassava starch paste properties

To investigate the natural variation of known genes related to starch metabolism and the possible molecular mechanisms underlying SNPs detected for the starch paste properties, the physical positions of the molecular markers were compared with the cassava reference genome deposited in the Phytozome version 12.1 database. The results demonstrated that two transcripts were the most likely candidate genes in the regulation of cassava starch paste properties: Manes.17G040300 [encoding glycosyl hydrolases family 35 enzyme—GH35—which hydrolyzes the glycosidic bond between two or more carbohydrates, or between a carbohydrate and a non-carbohydrate component [70], as well as a galactose-binding lectin domain, which is associated with chemical reactions and pathways involving carbohydrates] within the position 17302713 and 17311177 on chromosome 17; and Manes.18G041200 [encoding the mannose-p-dolichol utilization defect 1 (MPDU1) protein, which is involved in glycosyltransferase action] on chromosome 18.

GH35 is involved in chemical reactions and metabolic pathways associated with carbohydrates, or any group of organic compounds based on the general formula $Cx(H_2O)y$. It includes the formation of carbohydrate derivatives by adding carbohydrate residues to other molecules, through selective interaction with any carbohydrate, which includes monosaccharides, oligosaccharides, and polysaccharides [71]. On the other hand, MPDU1 has a biological function in glycosyl transferase, which consists of a superfamily containing hundreds of families that are generally located in the cytosol and are involved in the synthesis of products such as flavonoids, phenylpropanoids, terpenoids, and steroids, as well as plant hormonal regulation [72]. In addition, glycosyltransferases act in the antioxidative system, which is related to the neutralization and elimination of free radicals, thus preventing damage to biomolecules [73]. The glycosyltransferases, together with the glycosidases, comprise a group of enzymes of great importance in the biosynthesis and remodeling of the cell wall of carbohydrate polymers during different stages of plant development [74].

Other studies have reported that the genes GRMZM2G146028 (AP2/EREBP family transcription factor) on chromosome 4 and GRMZM2G142709 (glycosyltransferase) on chromosome 5 are likely candidate genes for QTLs qPV4-1 and qTV4-1, and qFV5-2, respectively, which regulate PeakVisc, trough viscosity, and FinalVis [57]. In other starchy species such as rice, a transcription factor of the AP2/EREBP family has been identified as a determinant in the physicochemical properties of starch, by regulating the starch biosynthesis underlying the amylopectin structure [75]. Based on the analysis of biparental cassava populations, [11] reported a QTL with greater effect for PastTemp (qPT.1LG1), colocalized with candidate genes encoding the family of glycosyl or glucosyl transferases and hydrolases at the periphery of QTL peaks. Therefore, these results corroborate our demonstration that GH35 and MPDU1 can be effectively associated with several RVA parameters of cassava starch. However, this effective association requires further study, which would involve strategies like the fine mapping of QTLs or knockouts and overexpression of candidate genes in QTL intervals. Other future approaches to connect the GWAS results to biological mechanisms involved in the cassava starch paste properties are transcriptomic studies aiming to identify genes in biological processes underlying complex traits. The integration of genomic variants and gene expression is of particular interest in cassava genomics of key traits for industrial uses.

Most of the other transcripts present in genomic regions associated with cassava starch paste properties have general functions in numerous cellular processes, including cell division, DNA duplex unwinding, endonucleolysis, endosperm development, protein folding, sorting and degradation, general metabolism, genetic information processing, glycerolipid metabolism, heme oxidation, photosynthesis, lipid metabolic processes, MAPKK activity, metabolism of cofactors and vitamins, negative regulation of the abscisic acid-activated signaling pathway and kinase activity, nucleic acid metabolic processes, phosphorylation, plant hormone signal transduction, protein phosphorylation, proteolysis, proton transmembrane transport, regulation of cyclin-dependent protein serine/threonine kinase activity, regulation of transcription, RNA phosphodiester bond hydrolysis, SCF-dependent proteasomal ubiquitin-dependent protein catabolic processes, signaling and cellular processes, and steroid biosynthetic processes [76].

This is one of the first studies using GWAS to understand cassava starch paste properties and can contribute to more complex investigations of variations in candidate genes that may affect different accessions, thus serving as a basis for gene cloning related to starch quality, in order to further understand its complex metabolic pathway and pasting properties. Future studies should focus on the resequencing the candidate genes and fine mapping of QTLs, with a large number of accessions and the validation of these candidate genes in different environments.

## 4.6 Final remarks

In this study, the genetic basis of cassava starch paste viscosity was dissected using GWAS with a medium to high density of SNPs markers. GWAS is a tool with enormous potential to accelerate breeding strategies to improve the quality of cassava roots for industrial use, as it allows breeders to make the selection based on the most effective marker-trait associations. In general, few genomic regions were associated with any RVA parameters on cassava chromosomes 3, 8, 17, and 18, indicating that a only a few genes may be responsible for the expression of these traits. Thus, the transfer of these genes/QTLs to different genetic backgrounds via backcrosses or even recurrent selection programs can be easily obtained and even monitored after the development and validation of specific markers for these RVA parameters.

This is a pioneering study on the genetic architecture and the mapping of genomic regions associated with cassava starch paste properties. Two SNPs located close to genomic regions that perform several functions in plant metabolism have been identified. Although they are not directly associated with the genes of starch biosynthesis, they are indicative of the discovery of new alleles that control these variations in cassava germplasm. In addition, as SNP markers are highly stable and can be genotyped in low-density genotyping service such as the KASP technology. Therefore, an important step to expand the use of our results will be the development of KASP assays to convert SNPs markers into breeder-friendly KASP markers, which are cost effective and usually offers rapid turnaround for trait screening and quality control to help breeders starch enhance pasting properties through marker-assisted selection.

## Supporting information

**S1 Fig. Regional marker-trait associations of the most significant loci for *setback* based on MLM, MLMM, and FarmCPU methods.** The middle layer shows the filtered genes and annotated sequences of the cassava genome (Phytozome v12.1), and the bottom layer shows the linkage disequilibrium matrix.
(TIF)

**S2 Fig. Regional marker-trait associations of the most significant loci for *PastTemp* based on FarmCPU method.** The middle layer shows the filtered genes and annotated sequences of the cassava genome (Phytozome v12.1) and the bottom layer shows the linkage disequilibrium matrix. (TIF)

**S3 Fig. Regional marker-trait associations of the most significant loci for *FinalVis* based on MLM, MLMM, and FarmCPU methods on chromosomes 17 and 18.** The middle layer shows the filtered genes and annotated sequences of the cassava genome (Phytozome v12.1) and the bottom layer shows the linkage disequilibrium matrix. (TIF)

**S4 Fig. Regional marker-trait associations of the most significant loci for *PeakVisc* based on MLM, MLMM, and FarmCPU methods on chromosomes 17 and 18.** The middle layer shows the filtered genes and annotated sequences of the cassava genome (Phytozome v12.1) and the bottom layer shows the linkage disequilibrium matrix. (TIF)

**S5 Fig. Regional marker-trait associations of the most significant loci for *Cold-PVisc* based on MLM, MLMM, and FarmCPU methods on chromosomes 17 and 18.** The middle layer shows the filtered genes and annotated sequences of the cassava genome (Phytozome v12.1) and the bottom layer shows the linkage disequilibrium matrix. (TIF)

**S6 Fig. Regional marker-trait associations of the most significant loci for *Hot-PVisc* based on MLM, MLMM, and FarmCPU methods on chromosomes 17 and 18.** The middle layer shows the filtered genes and annotated sequences of the cassava genome (Phytozome v12.1) and the bottom layer shows the linkage disequilibrium matrix. (TIF)

**S1 Table. List of cassava accessions with geographic and genetic origin, state of collection, and entry date in the Embrapa Cassava Germplasm Bank.** (DOCX)

**S2 Table. Average, maximum and minimum temperature and average rainfall by month during the crop season (2015/2016 and 2016/2017).** (DOCX)

**S3 Table. Genomic coverage of the single-nucleotide polymorphism (SNP) markers on the 18 cassava chromosomes.** (DOCX)

**S4 Table. Analysis summary of the PANTHER (PTH) and Pfam (PF) database, as well as EuKaryotic Orthologous Groups (KOG) and Gene Antology annotation.** For the SNPs, the list of annotated genes and transcripts close to each window explained most of the genetic variance for cassava starch pasting properties. (DOCX)

## Acknowledgments

The authors are grateful to the Conselho Nacional de Desenvolvimento Científico e Tecnológico, Fundação de Amparo à Pesquisa do Estado da Bahia, Coordenação de Aperfeiçoamento de Pessoal de Nível Superior for financial, and Nextgen Cassava Project for supporting the project and researchers.

## Author Contributions

**Conceptualization:** Cristiano Silva dos Santos, Massaine Bandeira Sousa, Eder Jorge de Oliveira.

**Data curation:** Cristiano Silva dos Santos, Massaine Bandeira Sousa, Ana Carla Brito, Luciana Alves de Oliveira, Carlos Wanderlei Piler Carvalho.

**Formal analysis:** Cristiano Silva dos Santos, Massaine Bandeira Sousa, Ana Carla Brito, Eder Jorge de Oliveira.

**Funding acquisition:** Eder Jorge de Oliveira.

**Investigation:** Cristiano Silva dos Santos, Massaine Bandeira Sousa, Ana Carla Brito.

**Methodology:** Cristiano Silva dos Santos, Massaine Bandeira Sousa, Ana Carla Brito.

**Project administration:** Eder Jorge de Oliveira.

**Resources:** Eder Jorge de Oliveira.

**Supervision:** Luciana Alves de Oliveira, Eder Jorge de Oliveira.

**Writing – original draft:** Cristiano Silva dos Santos, Massaine Bandeira Sousa, Ana Carla Brito.

**Writing – review & editing:** Luciana Alves de Oliveira, Carlos Wanderlei Piler Carvalho, Eder Jorge de Oliveira.

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
