## [Decision Letter · Decision Letter 0]

12 Oct 2021

PONE-D-21-19308Genome-wide association study of cassava starch paste propertiesPLOS ONE

Dear Dr. de Oliveira

Thank you for submitting your manuscript to PLOS ONE. After careful consideration, we feel that it has merit but does not fully meet PLOS ONE’s publication criteria as it currently stands. Therefore, we invite you to submit a revised version of the manuscript that addresses the points raised during the review process.

Please address the comments raised by reviewers and improve presentation of results and discussion section.Please ensure that your decision is justified on PLOS ONE’s publication criteria and not, for example, on novelty or perceived impact.

We look forward to receiving your revised manuscript.

Kind regards,

Harsh Raman, Ph.D

Academic Editor

PLOS ONE

Journal Requirements:

“The authors thank Conselho Nacional de Desenvolvimento Científico e Tecnológico, Fundação de Amparo à Pesquisa do Estado da Bahia, and Coordenação de Aperfeiçoamento de Pessoal de Nível Superior for financial and scholarship support. This work was also was partially supported by the NEXTGEN Cassava project, through a grant to Cornell University by the UK’s Foreign, Commonwealth & Development Office (FCDO) and the Bill & Melinda Gates Foundation (Grant INV-007637 http://www.gatesfoundation.org). “

“●           Eder Jorge de Oliveira: CNPq (Conselho Nacional de Desenvolvimento Científico e Tecnológico). Grant number: 409229/2018-0, 442050/2019-4 and 303912/2018-9

●             Eder Jorge de Oliveira: FAPESB (Fundação de Amparo à Pesquisa do Estado da Bahia). Grant number: Pronem 15/2014

●             Ana Carla Brito: FAPESB (Fundação de Amparo à Pesquisa do Estado da Bahia). Grant number: DCR0005/2015

●             Eder Jorge de Oliveira: UK’s Foreign, Commonwealth & Development Office (FCDO) and the Bill & Melinda Gates Foundation. Grant number: INV-007637. Under the grant conditions of the Foundation, a Creative Commons Attribution 4.0 Generic License has already been assigned to the Author Accepted Manuscript version that might arise from this submission.

●             The funder provided support in the form of fellowship and funds for the research, but did not have any additional role in the study design, data collection and analysis, decision to publish, or preparation of the manuscript.”

Additional Editor Comments (if provided):

Thanks for your patience. Overall paper is well written but requires some changes. Please revise it and resubmit. My comments are attached in the pdf.

Reviewers' comments:

Reviewer's Responses to Questions

**Comments to the Author**

1. Is the manuscript technically sound, and do the data support the conclusions?

Reviewer #1: Yes

2. Has the statistical analysis been performed appropriately and rigorously? 

Reviewer #1: Yes

3. Have the authors made all data underlying the findings in their manuscript fully available?

Reviewer #1: Yes

4. Is the manuscript presented in an intelligible fashion and written in standard English?

Reviewer #1: Yes

5. Review Comments to the Author

Reviewer #1: Title: Adequate

General comment: This manuscript utilized a diverse set of cassava population comprising landraces and improved cultivars from different countries for GWAS in the dissection of the genomic regions underlying cassava pasting properties based on different methods: MLM, MLMM and FarmCPU. To the best of the Authors’ knowledge, this is the first of such studies employing GWAS for mapping of such pasting property - related traits [pasting temperature, peak viscosity, hot-paste viscosity, cool-paste viscosity, final viscosity, breakdown, and setback] .in cassava. The present study is an important contribution in the field; the set of experiments and data analyses were well conducted; and the manuscript properly organized. I found this manuscript to be of sufficient merit and recommended for publication in PLOSE One after minor corrections.

I have made minor corrections and raised some comments and questions in the marked manuscript with tracked-changes which should be addressed by the authors.

Abstract: Ok.

Introduction: Ok. Clearly written and sufficient background of study provided

Materials and Methods: Adequate and well described

Results: Adequate. However, On S2 Table, Total and mean should be included at the bottom. This will aid readers to quickly know the Total size (Mb) of the 18 Chromosomes and also the total number of Markers (followed by the mean value per chromosome as you have already indicated). I could not find Table S1 among the supporting documents or in the main text (pdf).

Discussion: The results obtained were well discussed. Minor points were raised in a few aspects

Final Decision: The manuscript is of sufficient merit for Publication in PLOSE ONE. All the minor corrections and the additional comment should be taken into consideration.

I therefore recommended acceptance for publication with minor revision.

6. PLOS authors have the option to publish the peer review history of their article (what does this mean?). If published, this will include your full peer review and any attached files.

Reviewer #1: No

---

## [Author Response · Author response to Decision Letter 0]

5 Nov 2021

Reviewer #1: Comments: Word doc.

1) Line 91: Rephrase appropriately.

Response: Ok, we rephrased to “GWAS is a powerful method to identify..”

2) Line 175: Briefly indicate the reason for this modification “being the addition of polyvinylpyrrolidone (PVP) and increasing the concentration of 2-mercaptoethanol to 0.4%”.

Response: Ok, we complemented with “…, to break down organic material, especially tannins and phenolics”. 

3) Line 247: Do you consider this as low magnitude “amplitude values of SetBack (191.01 to 2,309.42 cP) and Hot-PVisc (351.41 to 2,692.87 cP)”. Why not refer to this variation as moderately high!

Response: We agreed and considered moderately high magnitude.

4) Line 248: How do you classify this magnitude of variation: High or moderate or low? 

Response: We considered low magnitude.

5) Line 261: … 0.66 to 0.69 is not low heritability. This is moderate heritability!

Response: Agreed.

6) Line 265: Look at Fig. 1 again. Should this not have rather been (0.80 < r <0.90) i.e. r is greater than 0.80 but less than 0.90?

Response: Corrected!

7) Line 273: On this S2 Table, Include Total and mean at the bottom. This will aid readers to quickly know the Total size (Mb) of the 18 Chromosomes and also the total number of Markers (followed by the mean value per chromosome as you have already indicated).

Response: We include total and mean in S2 Table.

8) Line 278: Table S1 and S1 Table? See S2 Table above. Be consistent with your captions. I cannot not find Table S1 among the supporting documents or in the main text (pdf).

Response: We correct to “S1 Table”.

9) Line 315: It might not necessarily be entirely due to being under the control of the same gene but could also perhaps be due to closely linked genes. Fine mapping approach will help to dissect the genetic architecture and clarify this observation. Elaborate on this point of fine mapping approach for future studies in the Discussion Section.

Response: Ok! We finish the discussion saying that other molecular approaches, as fine mapping, will be necessary to dissect the genetic architecture of starch pasting properties.

10) Line 341: Why do you think MLM was more powerful to detect these five QTLs and not the other methods?

Response: In general, the MLM, MLMM, and FarmCPU methods have different approaches to detect genomic regions associated with certain traits. The MLM model considered single-locus analysis and MLMM and FarmCPU multi-locus analysis. The single-locus models means that they comprise a one-dimensional genome scan by testing one marker at a time, iteratively for every marker in a dataset. This single-locus approach can also induce false negatives due to over fitting of the model where some potentially important associations can be missed. However, MLM was more powerful to detect SNPs in this study, than multi-locus approaches.

11) Line 356: In S1 Fig and S2 Fig,”annoted” should be corrected to “annotated”.

Response: Ok!

12) Line 453: See line 257. You were indicating these moderate heritability values as low. These values are moderate as you have indicated here.

Response: Ok!

13) Line 462: Are you referring to authors of (11)?. How would heritability values ever be 0.0. Does it mean there is no genetic contribution at al. Did they talk about epigenetics? But in your present study you obtained high heritability of 0.72 for what they reported a value of 0.0. The results by these authors is seemingly curious.

Response: The authors (11) attributed these low values to the strong environmental effect. We remove this sentence.

14) Line 542: Are you inferring these as minor QTLs?

Response: Yes!

15) Line 594: “of overlapping QTLs can control the same traits…“ is the point raised about fine mapping, and not necessarilly that these traits are under the control of the same gene. Add some information on how fine mapping can clarify this observation in future studies.

Response: Ok! We add the sentence “Fine-mapping is a strategy that can be used in future studies to determine the genetic variant responsible for pasting traits.” and we comment about fine mapping in the lines 701-707.

16) Line 688: It is good you mentioned fine mapping here. See my earlier comment on the overlapping QTLs.

Response: Ok!

Reviewer #2: Comments: PONE-D-21-19308.pdf

1) Line 38: why r2 at 0.1 was considered rather than 0.2?

Response: We consider that cassava has a medium size genome, and we see r2=0.1 is identified at several Mb distance. So, r2=0.1 is not a weak LD considering the distance. Moreover, we want to use same background level (r2 < 0.1) as used in another manuscripts in cassava such as https://doi.org/10.1007/s00122-021-03775-5

2) Line 44: Is it in %?

Response: No, this is the variance explained by the SNPs.

3) Line 61: pl define this “Starch retrogradation”.

Response: We include “Starch retrogradation” definition in line 64 “This process in which disaggregate the starch components, amylose and amylopectin chains, in a gelatinized starch paste and reassociate to form more ordered structures”.

4) Line 97: Why specifically used word natural? Have authors used material produced otherwise.

Response: We rewrite "to natural phenotypic variations of several traits" to "complex agronomic traits".

5) Line 126: define “Aw to Am according to the Köppen classification”?

Response: Ok! We rewrite to "Tropical Wet Savannah (Aw) to Tropical Monsoon (Am) according to the Köppen classification".

6) Line 129: Please provide plots of agro-climatic data (temps and rainfall of cropping season) as supplementary information

Response: Agreed. We added the S2 Table. 

7) Line 321 (Table 2: “MAF” is this essential? Suggest to delete). / R2 is in %? / What these values (-) mean. Pl detail it (line 322). / Add R2 values to MLMM and FarmCPU.

Response: Actions performed: 1) “MAF” column deleted; 2) R2 values range from 0 to 1 (not in %); 3) “-“ are spaces in the table without values. However, we remove “-“ to clean the table; 4) The R2 is only for single marker testing model. FarmCPU and MLMM are multiple markers testing models, so that the R2 value cannot be calculated.

8) Line 558: Any evidence to support “...it is possible to speculate that the traits Cold-PVisc, FinalVis, Hot-PVisc, and SetBack have additive effects due to the significant differences between the genotypes that have one or two copies of the favorable alleles…”.

Response: The copy number of the favorable alleles based on the SNPs S18_2907312 and S18_3567816 associated with Cold-PVisc, FinalVis, Hot-PVisc, and SetBack is based on the results observed in the boxplots (Figure 5). This result is explained as:

“The genotypes of SNP S17_17327003 exhibited significant differences for Cold-PVisc and FinalVis (i.e., the CG contents were approximately 379 and 329 cP higher than in the GG genotypes) (Fig 5). On the other hand, SNPs S18_2907312, S18_3081635, S18_3399799, S18_3399801, S18_3567791, and S18_3567816 showed that homozygous genotypes of favorable alleles (A, G, G, A, G, and T, respectively) had high Cold-PVisc, FinalVis, Hot-PVisc, and SetBack, while heterozygous and homozygous alternative alleles exhibited medium and low pasting property traits, respectively. The same trend was identified for SNP S18_3408138, in which the AA, AC, and CC genotypes had ~1,163 cP, 2,315 cP, and 2,937 cP, respectively, for Cold-PVisc, and 1,141 cP, 2,285 cP, and 2,890 cP, respectively, for FinalVis. These results indicated that the genetic control of Cold-PVisc, FinalVis, Hot-PVisc, and SetBack generally had additive effects. Therefore, increasing the frequency of those favorable alleles in the right direction would significantly improve the quality of cassava starch.”

9) “Therefore, it is possible that these correlated effects are primarily due to the genetic linkage instead of pleiotropy”: Any data to discount 'pleiotrophy'.

Response: No, just the evidence that we mentioned in the present study: “…moderate to high heritability, the large LD block on chromosome 18, and a few other regions associated with pasting traits, suggesting that a set of overlapping QTLs can control the same traits”.

10) Authors did not present any data on sequence variation among accession. There is no data on gene expression, I suggest limiting this to a small paragraph

Response: We added the following information on the Discussion section:

“However, this effective association requires further study, which would involve strategies like the fine mapping of QTLs or knockouts and overexpression of candidate genes in QTL intervals. Other future approaches to connect the GWAS results to biological mechanisms involved in the cassava starch paste properties are transcriptomic studies aiming to identify genes in biological processes underlying complex traits. The integration of genomic variants and gene expression is of particular interest in cassava genomics of key traits for industrial uses.”

11) In this paragraph, authors can talk about of genomic regions associated with cassava quality and how breeders can use this information (KASP etc)

Response: We added the following information on the Discussion section:

“In addition, as SNP markers are highly stable and can be genotyped in low-density genotyping service such as the KASP technology. Therefore, an important step to expand the use of our results will be the development of KASP assays to convert SNPs markers into breeder-friendly KASP markers, which are cost effective and usually offers rapid turnaround for trait screening and quality control to help breeders starch enhance pasting properties through marker-assisted selection.”

---

## [Decision Letter · Decision Letter 1]

10 Jan 2022

Genome-wide association study of cassava starch paste properties

PONE-D-21-19308R1

Dear Dr. Jorge de Oliveira,

We’re pleased to inform you that your manuscript has been judged scientifically suitable for publication and will be formally accepted for publication once it meets all outstanding technical requirements.

Kind regards,

Harsh Raman, Ph.D

Academic Editor

PLOS ONE

Additional Editor Comments (optional):

Reviewers' comments:

Reviewer's Responses to Questions

**Comments to the Author**

1. If the authors have adequately addressed your comments raised in a previous round of review and you feel that this manuscript is now acceptable for publication, you may indicate that here to bypass the “Comments to the Author” section, enter your conflict of interest statement in the “Confidential to Editor” section, and submit your "Accept" recommendation.

Reviewer #1: All comments have been addressed

2. Is the manuscript technically sound, and do the data support the conclusions?

Reviewer #1: Yes

3. Has the statistical analysis been performed appropriately and rigorously? 

Reviewer #1: Yes

4. Have the authors made all data underlying the findings in their manuscript fully available?

Reviewer #1: Yes

5. Is the manuscript presented in an intelligible fashion and written in standard English?

Reviewer #1: Yes

6. Review Comments to the Author

Reviewer #1: The revision is adequate. All the corrections pointed out have been effected by authors. I therefore recommend acceptance

7. PLOS authors have the option to publish the peer review history of their article (what does this mean?). If published, this will include your full peer review and any attached files.

Reviewer #1: **Yes: **Benjamin Ewa Ubi

---

## [Editor Report · Acceptance letter]

12 Jan 2022

PONE-D-21-19308R1 

Genome-wide association study of cassava starch paste properties 

Dear Dr. de Oliveira:

I'm pleased to inform you that your manuscript has been deemed suitable for publication in PLOS ONE. Congratulations! Your manuscript is now with our production department. 

Kind regards, 

on behalf of

Dr. Harsh Raman 

Academic Editor

PLOS ONE